# In Search of the Spin-Out Entrepreneur

Matteo Landoni [1,*] and dt ogilvie [2]

1   Department of Economic and Social History, School of Social and Political Sciences, University of Glasgow, Glasgow G12 8RT, UK

2   Saunders College of Business, Rochester Institute of Technology, Rochester, NY 14623, USA; dt@saunders.rit.edu

*   Correspondence: matteo.landoni@glasgow.ac.uk; Tel.: +44-(0)-141-330-5987

**Abstract:** A spin-out happens when an employee quits a company to start a new venture; however, theories do not agree on whether the 'spin-out entrepreneur' will start the company in the same or in a different industry. We investigated a sample of 250 entrepreneurs and 120 spin-out companies to understand what led an entrepreneur or a group of founders to enter a new industry. Our results contribute to theory, suggesting that spin-out entrepreneurs usually move to different and innovative industries owing to recombination of knowledge in founding teams. Our evidence supports the positive effect of different experiences within the team.

**Keywords:** spin-out; entrepreneurship; industry experience; entrepreneurial behaviour; knowledge flow; organisational limitations; founder

## 1. Introduction

'The garage is a bit of a myth' [1]. Contrary to popular belief, Apple co-founder Steve Wozniak admitted that the work behind the first Apple I in that summer of 1975 'was being done—soldering things together, putting the chips together, designing them, drawing them on drafting tables—at my cubicle at Hewlett–Packard (HP) in Cupertino'. Steve 'Woz' Wozniak, one of the founders of Apple computer together with iconic entrepreneur Steve Jobs and his colleague at Atari, the 'forgotten founder' Robert Wayne [2], contradicted the evocative 'image of the lone individual who relies primarily on his or her extraordinary efforts and talent' [3] (p. 7). This is not a speculation about the rhetoric behind the 'garage myth,' rather about the fact that often new ventures start while the founders are working in other organisations.

Deriving from this premise, our paper aims to shed light on how employees from existing companies start new ventures. We research questions relate to the practice of entrepreneuring—creating a new organisation that moves beyond presently dominant institutional arrangements [4]—by employees of an incumbent organisation. More specifically, we aim to disentangle the tension between the incumbent and the emerging organisations. Our research question is whether it is significant to start a new venture with or without the parent organisation consent.

First, we explore the theory in search of gap in this specific issue, and thus, we highlight the reason why this theoretical gap is not irrelevant and is otherwise worth of investigation. Second, we rely on illustrative cases that support our call for a deeper understanding of a defining feature.

Our paper examines the process of spin-out venturing with the aim to understand how much former employees replicate the same practices of their former organisation or move further and choose not to replicate those practices. For example, in terms of industry selection, do they enter the same, a related or a distant industry? In the case of non-replication, it is important to understand what characteristics of the founders affect the adoption of practices from previous work experience [5] (p. 20).

Our effort joins an already well-developed path of research that explores the impact of the characteristics of spin-outs on performance [6–11], and on their parent company's performance [12–15]. With the aim to enlarge the theoretical understanding of spin-out formation, we look at individual characteristics that can lead to spin-out occurrence and either positive or adverse effects for the parent firm [16–19].

This research contributes to the theory of the entrepreneur by exploring the characteristics that lead to the exploitation of work experience in a novel organisation, namely a spin-out. We define a spin-out as a new business venture that has no formal connection with the founders' previous organisations. This is because differences in cognitive styles between employees and managers can affect entrepreneurial choices and are closely connected to workplace behaviours [20].

The rest of the paper unfolds as follows. Section 2 defines spin-out and the theoretical framework of this research. Section 3 explores the gaps and develops the research questions. Section 4 selects an explorative sample and performs the analysis. Section 5 reports the results of our exploration. Eventually, Section 6 discusses the findings and concludes with suggestions for further research.

## 2. Research Background

### 2.1. Defining Spin-Outs

Spin-outs are new ventures founded by an employee of an existing company [21]. However, to spin-off or to spin-out is not the same. People indiscriminately call new businesses started by former employees either 'spin-offs' or 'spin-outs' [22] (p. 120, n1). However, for the purpose of this research, we use the term 'spin-out' to refer to a firm formed when an employee or group of employees leaves an existing entity to form an independent start-up firm without any formal linkage with the parent company. A 'spin-off,' on the other hand, happens when, as a deliberate act of strategy [23], a company makes a unit of itself an independent business [24] and keeps formal or informal linkages, such as equities, with the new company. Agarwal and colleagues [24] (p. 2, n1) declared that 'the terminology related to this organisational form is often confusing' and suggested a strict definition of spin–outs 'as an entrepreneurial venture founded by an employee of an incumbent firm within one year of leaving the incumbent and which competes in the same industry as the parent' [24] (p. 2).

Although firms have advantages, such as their scale, scope, tax or information, that could enable them to be more profitable than an employee in spinning off a discovery, an employee may exploit information asymmetries to start a business. Because tacit knowledge cannot be easily transferred [25], it makes it difficult for the employee to realise the actual value of the discovery from the company [24] (p. 2). As a result, there are three courses of action the employee can take: (1) Get the company to agree to a contract on the discovery before revealing the discovery to the employer; (2) develop a contract with the company after revealing the discovery; or (3) not reveal the discovery to the company in favour of developing it [26].

### 2.2. Knowledge Transfer

An employee transfers to the new venture those resources or knowledge that the incumbent company is not able or willing to process. Classical contributions to the studies of management have made clear how the resource firms possess both enable and constrain growth [27]. That means that the new knowledge that turns into new capabilities can enable firms to undertake new activities only when that knowledge is similar enough to the knowledge the firms already possess so they can absorb it [28,29]. Schumpeter is mostly right when he affirmed that 'in general it is not the owner of stage-coaches who builds railways' [30] (p. 66); nevertheless, an employee can acquire enough knowledge and leave without any approval to go beyond the cognitive limits [31,32] affecting owners and managers [33]. The cognitive distances among employees and employers as well

as among new venture founders deserve more research for their implications for open innovation [34].

'Pure knowledge spillovers' only happen if all the knowledge from employees as they move from employer to employer [35] are collected in an accessible common pool of knowledge [36]. Knowledge spillovers can lead to employee entrepreneurship because of the opportunities they uncover. An employee may see that her knowledge leads to a discovery and/or a business opportunity. When this happens, the employee may decide to start a new venture—a spin-out—either solo or with other employees or entrepreneurs, rather than stay with the firm [37]. Employees take their knowledge to the new venture. This is particularly true in skill-intensive industries, where the relationship between knowledge spillovers and the formation of new firms is strongest [38]. The knowledge of a firm comprises the knowledge of and in its employees and is developed through their actions [39]. Many studies argue that most entrepreneurs make use of business ideas encountered through previous employment [15,40–43]. Both empirical evidence and entrepreneurs' statements support this view.

A firm's knowledge generally resides in its people and develops through the efforts of employees [39]. When an employee starts or joins a new firm, he or she takes the knowledge gained at the prior firm to the new venture. In that case, an employee may have developed new knowledge that could lead her to a discovery and/or a business opportunity. Many studies argue that most entrepreneurs make use of business ideas encountered through previous employment [15,40–43]. Both empirical evidence and entrepreneurs' statements support this view.

Employment is a way to knowledge accumulation. Especially in the most innovative industries, employees often accept a lower wage in exchange for the opportunity to gain knowledge with the goal to start a business. They invest in the development of their human capital through the knowledge they will acquire as an employee by accepting a low wage [43]. Other scholars suggest that employees that start a venture while employed (hybrid entrepreneurship) decrease exit hazard and increase the survival rate of the spin-out company [44].

Incumbent firms are, in this sense, an important driving force of entrepreneurship. In fact, entrepreneurship can occur within an existing organisation [45]. Scholars have found that entrepreneurs have acquired useful knowledge about technologies [15,40], markets [46] and organisational processes [47] during their employment. Engel and colleagues [48] consider careers as a vehicle of experience accumulation that contributes to the development of effectual thinking, entrepreneurs' preference for a dominant decision-making logic.

*2.3. Employee Entrepreneurship and Industry Selection*

Employee-entrepreneurs may leverage innovations that the incumbent companies are slow to pursue because of either poor fit with their strategies or organisational limitations. Because many entrepreneurs conceive business ideas while working in established organisations [49,50], spin-outs represent a relevant case of start-ups [51]. More recently, scholars have agreed on a definition that necessitates the spin-out belonging to the same industry as the parent company and underscores the role of founders as conduits of knowledge from the parent firm to the new venture [11,15,24].

Although we agree on the distinction of employee entrepreneurship and the transfer of knowledge from the parent to the new venture, we feel the definition that bounds spin-outs in the same industry of the parent company is short-sighted. Indeed, this research seeks to find evidence of in which industries the spin-outs are likely to occur: the same industry of the parent company; a related business, such as a submarket; or in an unrelated business. In the latter case, it can be an optional use or an alternative use in a downstream market or submarket of the parent company's product or service by the new entrepreneurial firm. We define a submarket as an island of activity apart from the demand and supply side of the industry [52]. Spin-outs may appear as truly innovative firms; they can enter the submarket because they have lower costs and thus need less market share to be profitable

than other venture, and because they produce products similar to but differentiated from those of their parent companies [43].

Theory disputes the nature of spin-outs. On the one hand, some predict that employees will take advantage of their work experience, and thus spin-outs will act like their parents and select the same industry [14,40]. This means that founders are likely to identify entrepreneurial opportunities in the same industry of their former employers [17,35,53–55], or in a vertically related industry [56]. On the other hand, other scholars imply that spin-outs will not select the same industry of their parents because they found incompatible opportunities [43]. In that case, an employee can either start a new venture in a new industry as a *de novo entrant* or enter an industry related to the parent's industry [56]. In either case, the spin-out results from the experience of employees with the parent companies' technologies to the point of adapting such experience to a different industry [15]. Note that referring to a spin-out as the *generation* of a *parent* company resonates with the language of *procreation* and *heredity* that scholars widely imply as an explanatory framework for the *birth* of spin-outs [57].

The creation of entrepreneurial firms is only part of the reason for the growth of emergent industries. In fact, incumbent firms might also take part in the evolution of a new or emergent industry. Scholars distinguish the different factors that cause entry as either an innovative entrepreneurial (*de novo*) firm or a diversifying incumbent (*de alio*) firm [58]. Their roles in industrial emergence may lead them to differ in their strategies [59–61], timing [62,63] and performance [64,65]. Firms that enter an emergent industry by diversification (*de alio*) experience a benefit in the short run because of the amount of resources and capabilities available [61,63]. However, they progressively tend to lack adaptability to a new, dynamic environment where, in contrast, entrepreneurial firms (*de novo*) prove to be more innovative [66–68]. Either in the same or in a different industry, employees' experience seems to be the triggering factor of spin-outs.

## 3. Looking for the Spin-Out Entrepreneur

### 3.1. Transfer of Knowledge within the Same Industry

In the case of Apple at the start of this paper, the garage works similarly to a founding myth in present-day entrepreneurial society [69]. Actually, Wozniak kept working for HP several months after the founders formally filed the partnership papers for Apple Computer Company in 1976. Audia and Rider pointed out how the garage myth discounts the role of 'prior organisations in providing Jobs and Wozniak with confidence, exposure to fine-grained information, knowledge of the business and access to key social ties' [3] (p. 17).

This contradicts Schumpeter's heroic view of the entrepreneur [30,70] as a 'lone individual who relies primarily on his/her extraordinary efforts and talent to overcome the difficulties inherent in creating a new business' [3] (p. 19). More recently, entrepreneurial start-ups are similar to the story of the 'Traitorous Eight,' who were eight scientists who quit Shockley Semiconductor Laboratory in 1957 because they were dissatisfied and who went on to founding Fairchild Semiconductor. These eight scientists had been recruited a few years earlier by the Nobel Prize-winning William Shockley, one of the inventors of the transistor. They later became leaders of the semiconductor industry after leaving Shockley's laboratory [71–73], and founding first Fairchild Semiconductor and then Intel. Shockley considered their leaving a betrayal [3] (p. 19).

The story of these first pioneers of Silicon Valley soon became known worldwide as serial entrepreneurship [74], and paved the way for the study of inventors in the semiconductor industry [75]. The 'betrayal' was the 'ungrateful' transfer of knowledge from the existing company to a new venture. The stories of Apple and Fairchild depict two different eras and entrepreneurial ethos, but they both make us reflect on the importance of entrepreneurs' experiences as employees.

Spin-outs are a paradigmatic case of knowledge diffusion across organisations through employees [76]. However, the unauthorised transfer of knowledge from an incumbent to a spin-out underlies the agency costs of monitoring paid in companies. Incumbents

are most of the time not aware of the rate of knowledge appropriation and its conversion into an employees' entrepreneurial opportunity. Indeed, some scholars [14,77,78] consider spin-outs as a threat to the parent company because of the effect they may have on the ability of the incumbent firms to continue to compete in the market.

There is a certain amount of evidence suggesting that entrepreneurs inherit from their former employer technical and product knowledge [40,79–81], market-related knowhow [15,40], reputation and legitimacy [82], ties to customers and suppliers [83,84] and connections to financial and social capital [11,16,54]. It follows that the prior experience of entrepreneurs has a great influence on new ventures' performance [11,40,53,85,86]. The entrepreneur's prior employment is largely responsible for the growth and survival of new ventures [11,53,65,87–90].

**RQ1.** *Why and how spin-out entrepreneurs venture into the same industry?*

### 3.2. Entry into a New Industry

Other research studied how employee mobility affects knowledge flow. Knowledge flowing from one firm to another is a boost for the generation of new firms [91,92], and eventually to economic growth [93].

Innovations are the result of opportunities others did not see [94,95]. In this regard, scholars see spin-outs as one of the main sources of innovation [26,87,96–99]. The parent company may know the value of a discovery, but it cannot determine if any employee's discovery is valuable before she reveals it. There is a secret dimension to the knowledge the employee has accumulated, so if the employee does not require any parent company distinctive complementary assets she is more likely to start her business. R&D workers can best capitalise on their discoveries by starting a firm [14].

New businesses arise because of the innovations of entrepreneurs. High-tech scholars examine the diffusion of the spin-out phenomenon in local contexts such as Silicon Valley [100] and study new ventures that were started within existing organisations because of their importance in the diffusion of innovations. It should be noted that a spin-out firm's exploitation of an innovation does not necessarily threaten the parent company. For example, Klepper and Sleeper [15] found that managers in the U.S. medical devices industry did not have the necessary knowledge to evaluate and make informed decisions about opportunities that were unrelated to the core business of the firm [13,76].

A strategic commitment to the firm's core activity drives the management decision to dismiss employees' entrepreneurial opportunities [101] rather than limitations of organisational inefficiencies in diversifying [102].

Regarding industry structure, spin-outs are supposed to appear mainly in less mature industries [103]. In the more mature industries, innovation focuses more on production processes. Mature companies' know-how is more embedded in their physical assets than in their knowledge assets [104], leading to fewer opportunities for independent discovery from any new tacit knowledge of the employee. Moeen and Agarwal [105] call scholars' attention to how knowledge evolves during this nascent stage and how industry's knowledge evolution may subsequently reshape industry structure.

**RQ2.** *Why and how spin-out entrepreneurs venture into new industries?*

### 3.3. Methods: Industry Comparison

We compared the industry classification of spin-outs with their parent companies to determine whether the new venture was in the same or different industry as their parent companies. We adapted this method from Enkel and Gassmann [106], who used the European industrial classification code NACE (from the French *Nomenclature statistique des activités économiques dans la Communauté européenne*) to measure cross-industry innovation by evaluating the distance between industries. Echterhoff, Amshoff, Gausemeier [107] and Enkel and Heil [108] used this method for measuring knowledge heterogeneity in

cross-industry collaborations. Due to the great number of U.S.-based companies in our sample, we used the internationally adopted six-digit code from the 2012 *North American Industry Classification System* (NAICS), the standard code used by Federal statistical agencies in the United States, to identify industries.

The six-digit NAICS code designates the sector in the first two digits, and the subsector, group, local industry, and national industry in the third, fourth, fifth and the sixth digits respectively (BEA, 2016, www.bea.gov, accessed on 20 May 2022). The code for every company is public information, available online from the government agency, the U.S. Small Business Administration (www.sba.gov, accessed on 20 May 2022), or through online directories (for example, Manta, www.manta.com, accessed on 20 May 2022). We used the code to count the industry relatedness of the spin-out compared to the parent, considering them to be in the same business if six digits correspond, a sub-market if up to four digits correspond and a different industry when the first three digits differ. When the whole six-digit code differs, the spin-out operates in a completely different economic sector.

The NAICS code allowed us to count for the relatedness of the entrepreneurs' industry experience to the new business. Of course, in the case of a single founder venture, the industrial relatedness of the parent company and the entrepreneur's industry experience match. On the other hand, when a team of entrepreneurs starts a spin-out company, their industry experiences may differ. We measure the industrial relatedness of the spin-out by its difference from the NAICS code of the parent company, or the average difference for each entrepreneur when more parent companies are involved. With respect to this last case, the resulting mix of different industry experiences defines the heterogeneity of the founding team. By observing the composition of multiple-founder spin-outs, we assess the degree of heterogeneity of experience, first comparing the industry of origin of every entrepreneur, and second, by determining the average distance of each entrepreneur's parent company industry code from the spin-out's industry code. The resulting recombination of industry experiences may create a new set of knowledge combinations able to produce totally new and innovative business opportunities. That is, for example, the ability to move to a distinct market, especially in the sectors experiencing emerging technologies.

### 3.4. Sample and Data

We tested a sample of 250 entrepreneurs who started 120 spin-out firms. For the purpose of our research, we restricted our sample to cases in which entrepreneurs had relevant business experience in an established organisation—the parent company—that shared no formal ties with the new venture, neither having an ownership share nor deliberate involvement in the start of the spin-out. The spin-out entrepreneurs must have left the parent company without any commitment or approval.

It was not easy to select such a strictly defined subset of entrepreneurs. Nevertheless, an enduring effort lasting about two years allowed us to build a sample that possesses the desired features. We combed the web for data, first finding information about companies in business magazines and database websites (for instance, Forbes, entrepreneur.com (accessed on 20 May 2022), Crunchbase, etc.,) and second, collecting data about individual entrepreneurs on their corporate websites and by surfing their LinkedIn profiles. Thus, most of the data are public. We cautiously gathered the data and double-checked to ensure that they were accurate and trustworthy. This last process was quite long, but the result rewarded us with a brand new and unique dataset, original and internally consistent, that corresponds to the singularity of the case we wanted to test.

Despite the fact that our data cover many industries and a timespan of 60 years (1953–2013), half of the cases are no more than 15 years old. Around one-third of the cases are in the Information Technology (IT) and computer industries. The composition of the data provides further insights into the spin-out process itself, which industries are more involved with it, and how emerging technologies have shaped it.

## 4. Analysis

### 4.1. Descriptive Observations at the Firm Level

On average, there are slightly more than two entrepreneurs per company (2.08); 48 spin-outs are single founder companies, 39 have two founders, 22 have three founders, five have four founders, three have five founders, one has seven founders and two have eight founders (See Table 1).

**Table 1.** Number of entrepreneurs per spin-out company (N. spin-outs: 250).

| Entrepreneurs per Spin-Out | N. | % |
|:---:|:---:|:---:|
| 1 | 48 | 40 |
| 2 | 39 | 32.5 |
| 3 | 22 | 18.3 |
| 4 | 5 | 4.2 |
| 5 | 3 | 2.5 |
| 7 | 1 | 0.8 |
| 8 | 2 | 1.7 |
| Total | 120 | 100 |

Authors' elaboration.

The largest sector in the parent sample is Metal Manufacturing, which has slightly less than one-third of the companies (81 companies, 32.7%), followed by Professional, Scientific, and Technological Services (19.4%) and Information (8.9%).

The spin-out companies are in the sectors depicted by the first two digits of the NAICS code (See Table 2). The sample covers 16 different sectors with most occurring in Professional, Scientific and Technological Services, which has almost one-third of the companies observed (39 spin-outs, 32.5%); Information, which comprises the software and telecommunication industries (20 spin-outs: 16.7%); and Retail Trade (14 spin-outs, 11.7%).

**Table 2.** Spin-outs and the sectors of the parent companies.

| 2-Digit NAICS | Sector | Spin-Out | % | Parent Company | % |
|:---:|:---:|:---:|:---:|:---:|:---:|
| 21 | Mining, Oil and Gas | 1 | 0.8 | 0 | 0 |
| 22 | Utilities | 2 | 1.7 | 0 | 0 |
| 23 | Construction | 1 | 0.8 | 4 | 1.6 |
| 32 | Manufacturing (Except Metal) | 5 | 4.2 | 7 | 2.8 |
| 33 | Manufacturing (Incl. Metal) | 13 | 10.8 | 81 | 32.7 |
| 42 | Wholesale Trade | 5 | 4.1 | 9 | 3.6 |
| 44–45 | Retail Trade | 14 | 11.7 | 14 | 5.6 |
| 48 | Transportation | 3 | 2.5 | 2 | 0.8 |
| 51 | Information | 20 | 16.7 | 22 | 8.9 |
| 52 | Finance | 5 | 4.2 | 17 | 6.9 |
| 53 | Real Estate | 1 | 0.8 | 0 | 0 |
| 54 | Prof., sci, and tech. serv. | 39 | 32.5 | 48 | 19.4 |
| 56 | Administrative Services | 7 | 5.8 | 12 | 4.8 |
| 61 | Educational Services | 0 | 0 | 4 | 1.6 |
| 62 | Health Care | 0 | 0 | 3 | 1.2 |
| 72 | Accommodation and Food | 3 | 2.5 | 6 | 2.4 |
| 81 | Other Services (except Public) | 10 | 0.8 | 5 | 2.0 |
| 92 | Public Administration | 0 | 0 | 14 | 5.6 |
| Total | Valid | 120 | 100 | 248 | 99.2 |
| | Missing | *0* | *0* | *2* | *0.8* |

Authors' elaboration.

While the sectors are almost the same for parent companies and their spin-outs, the proportions are quite different. *Manufacturing* as a whole (either metal or not) accounts for 35.5% of the parents; however, the same sector is only 15% of the spin-outs. Conversely,

*Professional Services* jumps from 19.4% of the parents to 32.5% of the spin-outs; *Information* goes from 8.9% of parents to 16.7% of the spin-outs; and *Retail Trade* doubles from 5.6% of the parents to 11.7% of the spin-outs sample.

### 4.2. Outcomes at the Individual Level

We found a tendency for spin-outs to form in a different sector from the parent. We counted the distance between sectors, using as our measure the difference between the NAICS codes, or the average distance of digits of the parent companies' entrepreneurs in the case of multiple founders. In this industry classification system, a difference in the first two digits implies a different submarket in the same industry, a difference in digits three to five indicates a different industry and six digits of difference means a different sector. No difference in codes means, of course, that they are in the same business, market, industry and sector, which is only true for 11 spin-outs in the sample (9.2% of the total). Eleven spin-outs are in a submarket of the parent company, while 43 spin-outs (35.8%) appear in a different industry. The remaining 55 spin-out firms, accounting for the astonishingly high share of 45.8% of the total sample, operate in a completely different sector.

We explored the individual level to explain this. At least half of the entrepreneurs in every sector start their new venture in a different sector than their parent company. In aggregate, only 30% of the entrepreneurs do not move to a different sector, but two-thirds of these entrepreneurs entered a submarket. The entrepreneurs who spur more spin-outs in the same sector of origin as their parent companies are also in the larger sectors of the sample, *Manufacturing*, *Information* and *Professional, Scientific and Technological Services*; this last sector accounts for 45.8% of the cases, i.e., 22 entrepreneurs out of 48. Curiously, the same number of entrepreneurs from the *Manufacturing* sector started a *Professional Service* business.

The entrepreneurs' age at founding is 36.5 years old on average, ranging from 21 to 61 years old. There is a significant correlation between the entrepreneurs' ages at founding and the spin-outs' first year of activity: younger entrepreneurs started the more recent spin-outs (See Table 3). This result is in line with recent findings on hybrid entrepreneurship [109].

**Table 3.** Correlation.

| | | **1** | **2** | **3** | **4** | **5** |
|---|---|---|---|---|---|---|
| | | **Spin-Out Distance** | **Number of Entrepr.** | **Spin-Out Age** | **Age of Entrepr.** | **Age at Founding** |
| 1 | Corr. | 1 | | | | |
| | N. | (250) | | | | |
| 2 | Corr. | −0.28 ** | 1 | | | |
| | N. | (250) | (250) | | | |
| 3 | Corr. | 0.094 | 0.527 ** | 1 | | |
| | N. | (241) | (241) | (241) | | |
| 4 | Corr. | 0.11 | 0.484 ** | 0.867 ** | 1 | |
| | N. | (167) | (167) | (166) | (167) | |
| 5 | Corr. | −0.160 * | 0.159 * | 0.177 * | −0.34 ** | 1 |
| | N. | (167) | (167) | (166) | (167) | (167) |

* *p*-value 0.05; ** *p*-value 0.01 level (2-tailed test).

### 4.3. Outcomes at the Team Level

Looking at team composition, age at founding (M = 36.5 years, S.D. = 8.69) and the number of entrepreneurs per spin-out (M = 2.08, S.D. = 1.34) are associated (0.16. *p* = 0.04), but the correlation is in the opposite direction: the older the entrepreneurs, the larger the founding team. It follows that the older spin-outs tend to have more entrepreneurs in the founding team. Indeed, the age of the spin-out is highly and significantly correlated with the number of founding members (0.58. *p* ≤ 0.01). There is, however, no significant correlation between the year of the founding of the spin-out and the choice of a different

sector, industry, or market. In summary, the number of entrepreneurs on the founding team correlates to both the age of the spin-out as well as the age of entrepreneurs at the founding of the firm. The latter means that the more recent spin-outs, particularly the ones started after 2001, share smaller and younger founding teams compared to the older spin-outs.

Multiple-founder spin-outs are a peculiar case in that the entrepreneurs may have collected different industry experiences from the industries of the parent companies of origin. Thus, they can share different industry experiences among themselves and combine them within the team. In our sample, 202 entrepreneurs take part in 72 founding teams, for an average of 2.8 members on each team. In 40% of the teams, all of the entrepreneurs share experience in the same industry; in 45% of the remaining teams, the entrepreneurs have experiences not just in different downstream markets, but also in completely different sectors. The spin-outs that have higher heterogeneity within the founding team are in the *Information* sector, while the *Professional Services* sector, which is exactly one-third of the sample, shows no heterogeneity at all for half of the cases. The *Manufacturing* sector as a whole presents ambiguous results; however, when looking at its disaggregated components, the *Metal Manufacturing* spin-outs are greatly homogeneous, while the *Non-Metal Manufacturing* firms show high heterogeneity in all of the cases but one (See Table 4). Nevertheless, there are occurrences of single founders who succeed in venturing in a distant sector than their parent companies. Although the entry in a different industry by a single-founder spin-out happens slightly less often, more than half of the cases count a distance of at least five digits (see Table 5).

**Table 4.** Occurrences of heterogeneity in spin-out sectors.

| Code | Sectors | 0 | 2 | 3 | 4 | 5 | 6 | Tot | % |
|------|---------|---|---|---|---|---|---|-----|---|
| 21 | Mining, Oil and Gas | . | . | . | . | . | 1 | 1 | 1.39 |
| 22 | Utilities | 2 | . | . | . | . | . | 2 | 2.78 |
| 23 | Construction | . | . | . | . | 1 | . | 1 | 1.39 |
| 32 | Manufacturing (Ex. Metal) | 1 | . | . | . | 2 | 2 | 5 | 6.94 |
| 33 | Manufacturing (Metal) | 5 | 1 | . | . | 2 | . | 8 | 11.11 |
| 42 | Wholesale trade | 1 | . | . | . | 1 | . | 2 | 2.78 |
| 44–45 | Retail Trade | 2 | . | 1 | 1 | 2 | 1 | 7 | 9.72 |
| 48 | Transportation | . | . | . | . | 1 | . | 1 | 1.39 |
| 51 | Information | 4 | . | 1 | 2 | 6 | 2 | 15 | 20.83 |
| 52 | Finance | . | . | . | . | 2 | . | 2 | 2.78 |
| 53 | Real Estate | . | . | . | . | 1 | . | 1 | 1.39 |
| 54 | Prof., sci, and tech. serv. | 12 | 1 | 2 | 1 | 5 | 3 | 24 | 33.33 |
| 56 | Administrative services | 1 | . | . | . | . | . | 1 | 1.39 |
| 72 | Accommodation and Food | . | . | . | . | 1 | . | 1 | 1.39 |
| | Total | 29 | 2 | 4 | 4 | 24 | 9 | 72 | 100 |

Authors' elaboration.

**Table 5.** Founders' distance, single founders and teams.

| | Single-Founders | | Multiple-Founders | | Total | |
|---|---|---|---|---|---|---|
| Distance | N. | % | N. | % | N. | % |
| 0 | 5 | 10.4 | 6 | 8.3 | 11 | 9.2 |
| 2 | 2 | 4.2 | 2 | 2.8 | 4 | 3.4 |
| 3 | 4 | 8.3 | 3 | 4.2 | 7 | 5.8 |
| 4 | 5 | 10.4 | 6 | 8.3 | 11 | 9.2 |
| 5 | 14 | 29.2 | 19 | 26.4 | 33 | 27.0 |
| 6 | 18 | 37.5 | 36 | 50.0 | 54 | 45.0 |
| Total | 48 | 40 | 72 | 60 | 120 | 100 |

Authors' elaboration.

### 4.4. Heterogeneity in Teams

The number of founding team members is, partially counterintuitively, not a predictor of heterogeneity in the founding team. Others have predicted that the ability of the new venture to acquire knowledge increases with the size of the founding team [76,110]. On the contrary, our results show that the greater the number of team members, the higher is the occurrence of a homogeneous team; yet, only half of the 39 two-entrepreneur spin-outs mix different industry experiences. Therefore, a negative correlation exists between heterogeneity and the number of entrepreneurs ($-0.24$. $p = 0.043$).

The heterogeneity of the team explains the distance of the multi-founder spin-outs from the parent company industries. The internal distance among entrepreneurs' industrial experiences within the same team is highly and significantly correlated with the distance of the spin-outs from the average of the distances of each founder's parent company. As an example, consider PayPal. A team of five entrepreneurs started PayPal (NAICS code 522320), which was originally named Confinity. Besides serial entrepreneur Elon Musk [111], who was working at the time on a similar project (X.com), two of the entrepreneurs came from the financial services industry (CreditSuiss. NAICS code 522190), and two other members of the founding team came from the software industry (NetMeridian, NAICS code 334614; and Netscape, NAICS code 561499). The distance of the industry experiences of the five entrepreneurs is equal to the distance of the parent company code minus the spin-out company code. The values are 3, 3, 5, 6 and 5 (for example, $561499 - 522320 =$ a 5-digit distance), thus the mean value is 4.4. To clearly evaluate the distance separating the spin-outs from the parent companies, we rounded up all the results. It follows that the 'spin-out distance' is 5, corresponding to a different industry (see Table 6).

**Table 6.** The PayPal case.

| Entrepreneur | Parent Co. | Code | Indiv. Distance | Heterogeneity |
|---|---|---|---|---|
| Peter Thiel | CreditSuisse | 522190 | 130 | −4 |
| Ken Howery | Thiel Capital Mgmt | 522190 | 130 | −34 |
| Elon Musk | X.com | 511210 | 11,110 | −24 |
| Max Levchin | NetMeridian | 334614 | 187,706 | 6 |
| Luke Nosek | Netscape | 561499 | −39,179 | −5 |
| Digit | | 6 | 4.4 | 5.2 |

Note: PayPal Naics Code: 522320; Individual Distance = Parent-code—spinout-code; Heterogeneity = Parent-code—mean.

In this study, heterogeneity is the mean of the distance of the individual entrepreneurs to the spin-out distance. The average distance internal to the team of founders is 5.2 digits, which correspond to its heterogeneity value. We found that high heterogeneity is associated with high distance of industries.

The aggregated correlation of industries' distance and team heterogeneity is positively significant for the data in the sample (0.36. $p = 0.002$). Conversely, the spin-outs' distance from parents' industries is negatively correlated with the number of founders ($-0.465$. $p \leq 0.01$) and the age of the venture ($-0.308$. $p = 0.009$), which implies that the oldest spin-outs are less able to recombine different industry experiences and move into different industries or sectors than the newest spin-outs. At the same time, the smaller teams seem better able to manage heterogeneous experiences and pursue opportunities in different industries than the older, larger teams (See Table 7).

**Table 7.** Multiple-founder spin-outs correlation (m.: 72).

|  | **1** | **2** | **3** | **4** |
|---|---|---|---|---|
|  | **Spin-Out Distance** | **Spin-Outs Age** | **Number of Entrepreneurs** | **Team Heterogeneity** |
| 1 | 1 |  |  |  |
| 2 | −0.308 ** | 1 |  |  |
| 3 | −0.465 ** | 0.225 * | 1 |  |
| 4 | 0.360 ** | −0.226 | −0.239 * | 1 |

* *p*-value 0.05; ** *p*-value 0.01 level (2-tailed test).

## 5. Results

### 5.1. Innovation and the Industry Selection Tension

There are empirical studies on spin-outs including the research on Silicon Valley semiconductor producers from 1955–1981 [100]; U.S. commercial rigid disk drive producers from 1977–1997 [14]; and U.S. commercial laser producers from 1961–1994 [15]. The findings in these works suggest that spin-outs did not generally introduce significant innovations in their products and that their products are closely related to ones their parents produced. Founders of spin-outs commonly reported frustration with their parents as a major reason for leaving to start their own firms [87]. Some scholars [15] see spin-outs as a learning model of a differentiated product, pointing out that spin-outs will not use the same technologies of the parents to produce an identical product.

Companies are not the only ones suffering from organisational limitations in exploiting innovative discoveries. The employee who made the discovery can also find difficulties in turning it into a new business. In fact, many of the R&D workers employed in an organisation may not have the organisational skills needed to start a new firm [112].

### 5.2. Team and Experiences Recombination Tension

The case of the spin-out entrepreneur is a defined subset of entrepreneurial spawning that potentially contrasts with the existing literature, which assumes that founders rally teams that provide complementary human capital, where the focus is on complementarity of skills and tasks. Despite the fact that we showed that the complementarity of industry experiences within the founding team is relevant, it does not positively correlate with team size. In our sample, a larger number of entrepreneurs does not predict greater diversity (team heterogeneity); rather, it is the opposite. To team up with a co-founder who comes from a distant industry, as happens in the biotechnology industry, enables firms to reposition themselves technologically [113]. Some scholars argue that combining distant industry experiences may be too difficult to manage, so founders build teams of similar others in their new ventures [114,115]. Other scholars assert that social interaction augments knowledge flows between firms making the span of technological relatedness easier [116].

There is research that supports our findings about team size. When team members are similar and have homogeneous backgrounds their homophilous relationship breeds trust, thus lowering the cost of communication [117,118]. The trust that arises from homophily enables a larger team than would be possible to manage if the team were heterogeneous. Diverse teams are heterophilous, which means that smaller teams than possible with homogeneous teams would be mandated. Unlike homogeneous teams, small diverse teams require more face-to-face communications [119], which suggests that the cost of communication is lower, and its effectiveness is greater than if the heterogeneous teams were large. We can assume that when the industrial sector is unfamiliar, the spin-out companies in a novel industry primarily rely on the individual talent of a single entrepreneur or just a few.

## 6. Discussion

### 6.1. Spin-Out Entrepreneur

This paper explored the industry dynamics underlying the transfer of knowledge in the form of prior experience from an existing firm to a new venture. Unlike many others, we moved the unit of analysis from the perspective of the firm to the characteristics of individual entrepreneurs.

Our aim was to understand the mobility of employees as a crucial mechanism for the diffusion of knowledge, as suggested by other prominent scholars [120]. We contested that spin-outs select the same industry of their parents as suggested by learning theory [14]. Our results strongly support that spin-outs select a different industry, as many organisational scholars suggested [43]. In fact, this paper proposes not only that spin-out entrepreneurs tend to do something different, but we found that they pursue a new business in a different industry. That is even truer when there is a recombination of industry experiences. Individuals' judgments consider the cognitive distance between domains in the evaluation of business opportunities based on their experiences [121,122]. The resulting new knowledge leads to innovation in a new, emerging sector. Our results enlarge the perspective of organisational theory by adding insights to the recombination of prior experiences in innovative spin-outs.

We found that both mature businesses and innovative industries spawn spin-outs. Rather than comparing the number of new ventures, comparing the ability to move into other and often-emergent industries is a better measure of the innovative potential of spin-outs. In this case, the more traditional sectors (for example, manufacturing) prove to be less able to promote the recombination of knowledge.

### 6.2. The Relation between Spin-Out and Open Innovation

There seem to be some factors that allow spin-outs to convert submarkets into new disruptive technologies and industries. In this regard, the evidence of a strong and positive correlation between the heterogeneity of the founding team and the greater diversity of spin-outs from the parent companies' industries is by far an important contribution to the theory of entrepreneurship and innovation [105]. Despite the fact that the heterogeneity of industry experiences necessarily appears only in cases of multi-founder spin-outs, as well as that it is significantly associated with a greater ability to enter a different industry, heterogeneity is not a necessary condition for industry diversification.

The formation of spin-outs as result of an entrepreneurial action of an employee is relevant for its impact on the open innovation paradigm. The definition of open innovation is 'the use of purposive inflows and outflows of knowledge to accelerate internal innovation, and to expand the markets for external use of innovation, respectively' [123] (p. 1). Knowledge is central in the process of open innovation, and is the object of exploration, retention, and exploitation [124]. The editors of this special issue recall how the open innovation paradigm consists of companies opening to external sources of knowledge, but our research expands the paradigm to the counterintuitive and reverse direction of the company-to-innovation nexus. In our study, the company is not transferring knowledge from outside into its own innovation processes, but conversely it is the knowledge that exits the company that fulfils innovation through start-ups. The exploration of knowledge is not prerogative of firms, but of individuals in the company; the retention of knowledge in the firm depends on the opportunity for the individuals to bargain for its exploitation within their current company, and one—but certainly one of the most relevant to the paradigm of open innovation—is to transfer the knowledge outside the company in a new venture. The specific case of the spin-out entrepreneur is that of an open process of innovation leading to a closed outcome, i.e., the spin-out [125–127].

The case of outbound open innovation is neither new nor limited to spin-out entrepreneurship [128]; however, the emphasis on open-innovation primarily resides in inbound innovation, i.e., the flow of external knowledge into companies (id.). Our study helps to focus on the opposite event, the flow of knowledge to an organisation to a new

entrepreneurial venture. Furthermore, we inquiry the exploitation of the outflow of knowledge in new ventures, while most of the empirical studies on open innovation focus on large firms [129–131], and in multiple industries). The theory did not find a clear pattern on cross-industry processes of open innovation [125,132], and the reason is that the focus is mainly on the process within firms and much less on the external environment [130]; our research, instead, shifts the focus to multiple levels of analysis, spanning from the individual to the industry level and argues that the interplay of the different levels is essential in the process of knowledge interaction in the founding teams, as we expect that diversity leads to greater possibility for innovation, but also to greater cost of coordination. Nevertheless, a team-level perspective is increasingly regarded as a shaping force in the open innovation practices, for example is the field of open science [133].

## 7. Conclusions

Our contribution to open innovation theory is in refining the underdeveloped case of outbound open innovation, presenting the exploration of a well-defined case, the transfer of knowledge from a current organisation to a new venture, a case we define as spin-out entrepreneurship. We also contribute to the amount of empirical research on open innovation, changing the direction and the context of the process.

This paper acknowledges some limitations. The study suffers from the limits of a small sample that is the result of a trade-off between the availability of biographical data and the specificities of our research questions. To corroborate this result, it would be useful to carry out more research with both larger samples and different methods. Future research should, in particular, consider the role of entrepreneurial teams and the importance of collective dynamics. A new promising research avenue consists in applying a collective action perspective to entrepreneurship [134]. This research pointed out how the recombination of heterogeneous knowledge in founding teams spurs entrepreneurial endeavours, particularly in innovative fields of application. Innovation and entrepreneurship scholars are called upon to explore team practices and collective actions underlying the entrepreneurial process.

The main contribution of this study is in refining the definition of spin-out as the act to start a new venture while working for another company without any formal or informal agreement or involvement of the incumbent company. The spin-out entrepreneur, then, is the actor that transfer knowledge from old to new companies, and this usually happens in a different industry, producing inter-sectoral innovation and keeping an open communication from mature to emerging industry where knowledge can travel easily. This is relevant for the paradigm of open innovation. Despite the common sense definition of open innovation as the attraction, retention and transformation of external knowledge into companies, the case of companies spurring knowledge as entrepreneurial actions in the same or in a different industry exists, and it deserves more comprehension as a preferred way for the diffusion of innovation.

**Author Contributions:** Conceptualisation, M.L. and d.o.; methodology, M.L.; investigation, M.L.; resources, d.o.; data curation, M.L.; analysis, M.L.; writing—original draft preparation, M.L.; writing—review and editing, d.o.; supervision, d.o.; project administration, M.L. and d.o. All authors have read and agreed to the published version of the manuscript.

**Funding:** This research received no external funding.

**Institutional Review Board Statement:** Not applicable.

**Informed Consent Statement:** Not applicable.

**Data Availability Statement:** Not applicable.

**Acknowledgments:** We are thankful to The Center for Urban Entrepreneurship & Economic Development (CUEED) at Rutgers Business School in Newark, NJ, for research support.

**Conflicts of Interest:** The authors declare no conflict of interest.

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
