# Peer review of "In Search of the Spin-Out Entrepreneur"

_2199-8531, doi:10.3390/joitmc8030106_

Round 1

Reviewer 1 Report

First of all, I want to congratulate the authors for the good work done in this article.

In my opinion, it is a very relevant and current topic with important scientific repercussions. It is well written, with fluid and clear writing that makes reading easy and enjoyable.

The literature review reveals a lack of more recent references that make it difficult to frame the discussion in the present days.

The methodology is correctly developed and suitable for the purpose of the article. Discussion of results is a good bridge between current knowledge and the results obtained.

Although the positive aspects are by far more important, I would like to point out two significant weaknesses. First, the literature review, which I have already mentioned, should look for some current references given the current situation of the topic. The other aspect is related to the size of the sample, which prevents a more comprehensive and universal conclusion from being drawn.

Author Response

Thanks for your valuable comments, authors have revised the manuscript accordingly, please find the revised version in attachment.

Reviewer 2 Report

Interesting paper but still with some limitations regarding the results presented.  The authors refer exactly that in the conclusions, but my question is whether it would make more sense to publish the paper only when some of the limitations presented are already addressed. Although I consider that the article already presents some interesting data, it might be more interesting to publish the work when it is more complete.

Author Response

(The authors gave the same response as above.)

Reviewer 3 Report

I am glad that the authors took up the topic of spen-out enterprises. I highly appreciate this scientific work and congratulations to the authors. My doubts are only a suggestion for consideration by the authors.

1. I find the purpose of this paper very blurry. In the introduction, the goal is 'to shed light on how employees in existing companies start new ventures' (which was ambiguous), but in the following sentences the authors intend to do more. However, in the last chapter, the purpose is stated differently: 'Our aim was to understand the mobility of employees as a crucial mechanism for the diffusion of knowledge, ...'. Maybe it's worth putting it in order?

2. I appreciate the oratory of the authors, but there are too many colloquialisms and popular scientific terms in this article. Sometimes they are even difficult to understand. An example can be the first sentence: 'The garage is a bit of a myth' [1]. but also other statements: 'forgotten founder' 'garage myth,' etc. I propose to keep the scientific rhetoric.

Congratulations and good luck.

Author Response

(The authors gave the same response as above.)
